# Visual–Tactile Perception of Biobased Composites

**DOI:** 10.3390/ma16051844

**Published:** 2023-02-23

**Authors:** Manu Thundathil, Ali Reza Nazmi, Bahareh Shahri, Nick Emerson, Jörg Müssig, Tim Huber

**Affiliations:** 1School of Product Design, University of Canterbury, Christchurch 8041, New Zealand; 2The Biological Materials Group, Biomimetics, Faculty 5, HSB–City University of Applied Sciences Bremen, 28199 Bremen, Germany; 3Luxembourg Institute of Science and Technology, 4940 Hautcharage, Luxembourg

**Keywords:** biocomposite, material perception, aesthetics, material sustainability, product design, visual and tactile senses, material design, semantic differential method

## Abstract

Biobased composites offer unique properties in the context of sustainable material production as well as end-of-life disposal, which places them as viable alternatives to fossil-fuel-based materials. However, the large-scale application of these materials in product design is hindered by their perceptual handicaps and understanding the mechanism of biobased composite perception, and its constituents could pave the way to creating commercially successful biobased composites. This study examines the role of bimodal (visual and tactile) sensory evaluation in the formation of biobased composite perception through the Semantic Differential method. It is observed that the biobased composites could be grouped into different clusters based on the dominance and interplay of various senses in perception forming. Attributes such as *Natural*, *Beautiful*, and *Valuable* are seen to correlate with each other positively and are influenced by both visual and tactile characteristics of the biobased composites. Attributes such as *Complex*, *Interesting*, and *Unusual* are also positively correlated but dominated by visual stimuli. The perceptual relationships and components of beauty, naturality, and value and their constituent attributes are identified, along with the visual and tactile characteristics that influence these assessments. Material design leveraging these biobased composite characteristics could lead to the creation of sustainable materials that would be more attractive to designers and consumers.

## 1. Introduction

Biobased composites provide an effective alternative to the use of fossil-fuel-based materials used in mass production. Such materials are particularly relevant in the field of consumer products, where the production volumes are high, and the products have a short–medium lifespan [1]. By virtue of their biobased components, these composites provide materials with better sustainability and reduce adverse environmental effects [2]. However, decades of research into biobased composites have failed to create materials that are widely accepted as a viable option, which has been proposed to be a product of their perceptual handicaps [3]. Biobased composites are often considered to have poor aesthetics [4,5], presumably due to their typically fibrous, brownish, uneven visual characteristics and lack of aesthetic options available in fossil-based composites. These materials are also considered to be of lower value than other materials and fail to clearly demonstrate the naturality of their constituent parts. These perceptual handicaps could be classified as (i) lack of desirability, which constitutes unpleasant aesthetics and low perception of value, and (ii) lack of naturality, which is the core identifier for biobased composites to differentiate themselves from conventional composites. To create biobased composites with better desirability and naturality, the components of the perception-forming mechanism in biobased composites need to be examined, with a specific focus on senses (vision and tactility) and emotional attributes formed against material characteristics. The effect of other sensory signals (auditory, olfactory, and gustatory) are minor in the context of product design and hence are not included in this study.

Some have considered vision the primary sense for making sense of the world around us and the most important one [6,7,8]. In the context of the perception of biobased composites, the sensory inputs are primarily haptics and vision, with the influence of other senses limited [9]. The absence of either of these senses can impact material perception significantly. The loss of familiarity with materials and products in a nontactile scenario, as described by Schifferstein and Desmet [10], is attributed as one reason consumers prefer retailers offering a tactile examination of products before purchase, especially fabrics [11]. Klatzky et al. [12] argue that while visual stimuli help with the perception of the product form, tactility is the major contributor to material assessment and perception. These observations are supported by studies that noted higher emotional engagement and product recognition for consumers having physical product experiences [13,14]. These varying levels of impact of visual versus tactile sensations are dependent on the material or product category; Marlow and Jansson-Boyd [7] observe that while tactile sensations are dominant in forming product perception for materials such as fabrics, vision governs the constitution of perception in fast-moving consumer goods (FMCGs).

However, it may be argued that the sensory inputs one receives from the material are not distinct sensory signals and may be in the confluence of various sensory categories. This is supported by the concept of the multimodal nature of our sensorimotor system, which posits that our brain areas tasked with visual and tactile processing are cross-linked through neural links [15,16,17,18]. This means that such factors are interlinked, and neither tactile nor visual stimuli alone can determine material perception, pointing to a bimodal mechanism in the case of biobased composites. This interdependence was evident in the study by Dagman et al. [19], where participants anticipated tactile impressions based on vision and vice versa. Another aspect of sensory perception is that even one sense could include signals from various ranges within that individual sensory spectrum. For example, Okamoto et al. [20] have subdivided material tactility into five subdimensions: surface friction, temperature, hardness/smoothness, macro roughness, and fine roughness, illustrating the complexity of tactile sensations. 

The dominance of these sensory inputs and their interplay affects the formation of various attribute characteristics while perceiving materials. Certain material characteristics elicit corresponding emotional attributes in observers, leading to the formation of a material character. Some anecdotal evidence exists of the correlation between surface roughness and material warmth with naturality [21,22,23]. It is observed that people consider smooth surfaces to be more pleasurable than rough ones [24] and that perception of smoothness correlates with beauty, strength, and value in biobased composites [22]. While it is seen here that there are effects of tactile characteristics on material perception, the specific effects are still being uncovered. Understanding these effects is crucial for developing new sustainable materials, such as biobased composites, which could substitute fossil-fuel-based materials in product design and manufacture. Next to studying tactility and vision individually in a unimodal approach, it is necessary to explore the bimodal (visual–tactile) interaction of stimuli on material perception as this is the default setting for most product–user interactions. Therefore, in this study, we consider bimodal (visual–tactile) interactions in the context of biobased composites, with a focus on naturality and desirability (desirability expressed as a combination of beauty and value) to create tailored material experiences for users. 

This will be achieved by an examination of attribute–attribute correlations (e.g., are materials perceived as beautiful also perceived as valuable?) and material–attribute correlations (e.g., does the wood sample have high naturality ratings?) in biobased composites, focusing on the impact of combined visual–tactile senses on material perception. The primary objective is to reveal critical attributes that form a holistic material perception to identify material characteristics (visual and tactile) that correlate with attribute ratings. This exploratory study will help identify the material–attribute relationships in biobased composites so that new materials may be designed with desired perceptual qualities.

## 2. Materials and Methods

This study followed the methodology used for visual perception analysis and used the same 11 biobased material samples as detailed in [25] (Figure 1), which represented the types of common biobased materials used in manufacturing (random fibres, unidirectional fibres, woven fabrics, and natural materials). Among these materials, all biobased composites for this study were fabricated at the University of Canterbury, Christchurch, New Zealand (material details provided in the Appendix A). Since the focus of the study was on material perception, flat, rectangular forms of 50 mm × 35 mm were used to avoid the influence of form in perception [12,26]. Since samples differed in thickness, a depression was created on the cardboard base to avoid the influence of sample thickness in forming the perception. This mount also prevented participants from bending or breaking the samples to assess their strength. The natural materials used in this study (walnut wood, poplar wood, and leather) served as familiar reference samples to compare the naturality and desirability assessments of biobased composites. None of the samples exhibited discernable olfactory characteristics that influenced perceptual assessments.

A total of 51 adult participants were randomly selected for this study from the staff and students at the University of Canterbury, Christchurch, New Zealand. The 51 participants of this study included 29 female and 22 male participants. They represented 11 nationalities, with a median age of 22 (maximum—63, minimum—18), and had varied educational profiles. All participants had education beyond high school, and 35% possessed a postgraduation qualification or higher. This study did not accept subjects with impaired vision/colour blindness, neurological/psychiatric conditions impairing visual perception, or poor proficiency in English. The effect of demographics on biobased composite perception was not explored as prior studies failed to observe any significant effect of age or gender on perception [25,27]. 

For assessment, material samples were placed on a table, and participants were asked to rate each sample on a 5-point semantic differential scale (e.g., *Definitely Natural, Looks Like Natural, Can’t say, Looks Like Artificial*, and *Definitely Artificial*) with 10 bipolar attributes (Table 1) selected from prior literature [28,29]. The responses were recorded using QUALTRICS XM (https://www.qualtrics.com/), the online survey tool from QUALTRICS LLC (North Sydney, NSW, Australia), North Sydney, Australia (www.qualtrics.com). It should be noted that while there have been some concerns raised about the semantic differential method (ambiguity in adjective meanings, unequal intervals, ambiguity of middle-point assessments) [30], this method has been chosen here as the objective of this study is not to quantify the responses explicitly. The semantic differential method is applied here to examine the relative perceptual assessments of various samples and to extrapolate general material perception trends.

The ratings against each material–attribute combination were compiled to calculate the median ratings and also to identify the favourable rating percentages. This combined the frequency of *Definitely* and *Looks Like* ratings against each attribute, and the percentage of these favourable responses amongst all responses were calculated, e.g., Cellulose + Wood was rated *Definitely Natural* by 17 out of 51 participants and *Looks Like Natural* by 20, bringing the total to 37 favourable responses (72.5%) to *Natural*. Similarly, the percentage of ratings against *Can’t Say* revealed the uncertainty in perceptual assessment. Spearman’s rank correlation method was used to identify statistically significant attribute–attribute correlations (e.g., if ratings for *Natural* and *Valuable* correlated). Significant correlations were further grouped into similar attribute categories to generalise the perceptual relationships. In the assessments for the impact of tactility in ratings as well as for attribute–attribute correlations, the statistical tools used here employ multiple testing and face a higher chance of encountering type I errors. False detection rate (FDR) correction procedures are not employed for this study as (i) these may lead to higher type II error probability, and (ii) since the current study is exploratory, FDR techniques may limit the number of interactions between variables [31].

Since the material samples are analysed across 10 scales and 20 attributes, the t-distributed stochastic neighbour embedding (t-SNE) method was applied to reduce the multidimensional nature of the dataset to a two-dimensional one [32]. This was achieved through an open-source data visualisation software, ORANGE 3.30 (under GNU General Public License, https://orangedatamining.com/, accessed on 19 October 2022). This provided a clustered view of the materials in the study and helps to generalise visual/tactile characteristics and their perceptual implications.

## 3. Results and Discussion

### 3.1. Effect of Material Characteristics on Perception

To understand the effect of material characteristics on perception, it is essential to examine material–material relationships, and the t-SNE method provides the following clustering (Figure 2) for the biobased composite materials used in this study.

This classifies the group of 11 materials into four clusters. The common characteristics of Cluster 1 materials are that they have smooth resin surfaces and no fibre presence. Here, NW Sisal had a markedly different visual appearance versus the other two materials, and it may be assumed that tactile characteristics were the unifying feature for materials in this cluster. 

Cluster 2 materials (Poplar, Walnut, and Leather) are all non-composite natural materials but with different tactile features (wood grains, leather texture), pointing to the dominance of visual stimuli. Cluster 3 materials were perceived as the roughest materials; with fibres present on the surface, these materials had a visually divergent appearance from each other but similar tactile characteristics. Cluster 4 materials provide a mix of both the senses; these are neither visually nor tactilely congruent. While NW Coir is smooth, the other two materials had a fine roughness; while UD flax had a unidirectional orientation of fibres, the other two had randomly oriented fibres. This perceptual clustering of these materials may result from bimodal perception, where both visual and tactile senses exert equal influence.

### 3.2. Attribute–Attribute Correlations

Figure 3 visualises the correlations between various attributes and their relative strengths. These can be clustered into two closely correlated groups (correlated attributes with high Spearman’s Rho values and a high number of mutual correlations), the first of which consist of *Natural–Artificial, Beautiful–Ugly*, and *Valuable–Worthless* scales (Table 1), which have both visual and tactile influences. The second group consists of *Complex–Simple, Interesting–Boring, Unusual–Ordinary*, and *New–Aged* attribute scales. As illustrated in Figure 3, these two groups are connected through a set of intermediary attributes such as *Rough–Smooth, Strong–Weak*, and *Hot–Cold*, which are tactilely dominant. This also implies that the key group of attributes (beauty, naturality, and value) are closely interrelated in a bimodal context. Additionally, no strong, direct relationships exist between these key and visually dominant attributes, pointing to a clear bimodal influence in perception formation. This could also mean that materials require bimodal interaction to assess beauty, naturality, and value, and purely visual (unimodal) assessment may be erroneous. This also illustrates that visually dominant attributes such as complexity, unusualness, and interestingness have limited influence on the core objectives of this study (beauty, naturalness and value), and the impact of these attributes is limited to the tactile perception of materials.

In the *Complex–Simple* scale, all three natural materials (Cluster 2) were most clearly rated as Simple compared to the other samples. *Complex* also positively correlated with *Interesting* and *Unusual* and negatively with Boring, Ordinary, and Smooth, i.e., the more complex, the more interesting and unusual, and the less boring or smooth. It has been noted that high complexity leads to novelty [33], and this argument is validated here. The complexity scale also had 11 statistically significant correlations with other attributes, indicating it to be a critical component in perception forming. Cluster 4 materials (NW Coir, Cellulose + Wood, and UD Flax) were rated highly on the *Interesting–Complex* scale, which may suggest the influence of bimodal stimuli and the interplay between the two senses. Cluster 2 (natural materials) was ranked as the most *Simple* materials, which may imply that the participants were able to process the stimuli from these materials with ease cognitively. This aligns with the argument of Sharan et al. [34], who suggested that natural materials are easy to distinguish.

Cellulose + Wood, NW Coir, and UD Flax (Cluster 4) had the highest percentages of ratings on the *Interesting* side of the scale (combining *Definitely* and *Looks like*), and NW Coir and UD Flax were rated as *Definitely Interesting*. These Cluster 4 materials do not possess any common visual or tactile characteristics and hence are aggregated based on bimodal perception. The natural materials (Cluster 2) with visual aspects dominating perception were ranked less interesting than Cluster 4 materials. *Interesting* had significant positive correlations with *Complex* and *Unusual*, which were reinforced with the inverse correlations with their antonymous attributes. Though an almost linear trend exists between *Unusual* and *Interesting* ratings (Appendix A), Walnut, Leather, NW Sisal, and Rayon do not follow this. This reduction in unusualness may be attributed to the ease of visual processing in natural materials derived from familiarity [35], combined with visual dominance in the perception of Cluster 2 materials. In the case of NW Sisal and Rayon, the high levels of unusualness may be attributed to the visual–tactile conflict where both materials have high fibreness, but unique tactility (rough–soft or smooth–hard). Examples of such experiences could be woollen fabrics, which have a rough, uneven surface that is soft to stroke, or a highly polished wood surface, which is smooth to stroke but hard as well.

Most of the materials (7 out of 11) were rated as natural, with Walnut, UD Flax, and Poplar being rated as *Definitely Natural* (Figure 4). Strong positive correlations were observed between *Natural* and other attributes such as *Beautiful* and *Valuable*; these were bolstered by the negative correlations with *Ugly* and *Worthless*. The overall high ratings for the beauty and value of natural materials are also seen in earlier literature [36]. *Natural* also had a negative correlation with *Cold*, which was reported in prior literature [21].

Cluster 2 materials (Walnut, Poplar, and Leather) were rated highly for both beauty and naturalness (Figure 5), while Cluster 1 materials (TW Cotton, TW Flax 1, and NW Sisal) were rated low on both these scales. Labbe et al. [37], in their study of fabrics and papers, found touch to be central to the formation of perception, but that does not seem to be the case here as biobased composites like UD flax/Cellulose + Wood were rated highly for naturality. However, they were tactilely different from the natural materials in Cluster 2, based on the tactility classification by Okamoto et al. [20].

In the *Unusual–Ordinary* scale, NW Coir was rated as *Definitely Unusual*, owing to its discordant sensory characteristics (visually rough–tactilely smooth). Poplar was rated as *Definitely Ordinary,* presumably due to its lack of significant visual or tactile features. All the natural materials (Cluster 2) were rated as *Ordinary*. *Unusual* had positive correlations with *Complex* and *Interesting* and negative correlations with antonymous attributes in these scales (Figure 3); this relationship has also been observed in the visual perception study [25]. A plot of *Unusual* versus *Simple* (Appendix A) illustrates the natural materials (Cluster 2) ranking low on the *Unusual* scale, with Cluster 4 ranking highest. This low ranking may be attributed to both stimuli channels; visually simple interface as well as tactilely simple (smooth) surfaces.

In the *Beautiful–Ugly* scale, Cluster 2 materials were most clearly rated as beautiful (Figure 6), while Rayon, NW Sisal, TW Cotton, and TW Flax 1 (Cluster 1 materials) were rated as *Looks like Ugly*. This high sensitivity to natural patterns and their beauty may be due to the evolutionary cognitive traits of human beings [34,38]. *Beautiful* had positive correlations with *Natural* and *Valuable* and negative correlations with antonymous adjectives. The correlation of *Beautiful* with *Valuable* may be because the human reward systems in the brain are activated by beautiful things [15]. It is interesting to note that while Leather scored lower than Walnut on the *Beautiful* scale, it was deemed almost equally valuable; this points to perceptual mechanisms for the value being dependent on more variables beyond beauty, such as functional value or value associated with the usefulness of the object. NW Coir is also seen rated as the most beautiful among the biobased composite samples after natural materials. This is due to the fibrous yet smooth surface of this material, which has been shown to be perceived as beautiful and valuable [39].

Only Walnut and Leather were rated as *Valuable*; natural materials (Cluster 2) were positioned as the most valuable materials in the group, while NW Sisal, Rayon, and TW Flax 2 were rated the most *Worthless* materials (Figure 7). This scale had relatively high uncertainty in responses (23.7% *Can’t Say* responses) but had ten statistically significant correlations associated with other attributes (Appendix A) revealing the importance and assessment difficulty of this scale. *Valuable* had positive correlations with *Smooth, Natural*, and *Beautiful* and negative correlations among their antonymous adjectives. This is in line with assertions that consumers assign beautiful objects more value [40,41]. Natural materials (Cluster 2) were rated *Smooth* and *Valuable*, whereas Rayon and TW Flax 2 (Cluster 3) were rated *Rough* and *Worthless*. The same trend observed for NW Coir in the *Beautiful* scale is seen here for value as well, which aligns with previous research [39].

In general, the perception of smoothness correlates strongly with the perception of strength, while materials with rough surfaces and exposed fibres on their surface resulted in the tactile perception of softness and were rated the weakest amongst the samples. Walnut, followed by TW Flax 1 and NW Coir, had the highest ratings for strength, while Rayon, TW Flax 2, and UD Flax were rated as the weakest. This is interesting as unidirectional fibre reinforcement (as in UD Flax) would have had the highest strength and stiffness values, but this fact remains unclear to a casual observer. Cluster 2 (natural materials) was rated highly on *Smooth*, but the variance within the group for strength is significant, indicating the influence of visual perception in the assessment of strength. The impact of the *Rough–Smooth* scale is evident as it is one of the strongest tactile identifiers for material perception; a previous study by Karlsson and Velasco [42] considers this attribute the ‘easiest to judge’. A large intracluster variance is observed in smoothness, along with a clear distinction between smooth and rough materials in the group. This scale gave strong assessments, with very low uncertainty (*Can’t Say*—3.1%) and 8 out of the 11 materials being strongly associated with either extreme of the scale (*Definitely Rough* or *Definitely Smooth*). TW Flax 2, Rayon (Cluster 3), and UD Flax were rated the roughest, and Leather, Walnut, and Poplar (natural materials, Cluster 2) were rated as the smoothest materials.

*Smooth* had positive correlations with *Valuable, Strong*, and *Simple* and negative correlations with *Complex* and *Weak*. This relationship between smoothness and the perceived value of materials may be because people find smooth textures more pleasant than rough surfaces [6]; this perceived pleasantness may be analogous to material valuation. At the same time, it is also worth noting the variance of *Rough–Smooth* ratings with respect to valuation; this may point towards the inability of participants to differentiate macro roughness (surface unevenness larger than 1 mm) from fine roughness (surface unevenness below 1 mm, in µm range), as suggested by Okamoto [20]. This inability may have forced participants to choose an extreme (smooth/rough) and may explain the prevalence of extreme ratings for this scale. Karlsson and Velasco [42] found conflicting assessments for *Rough–Smooth* in the context of products; it is unclear if such an effect is present in the current study. 

The *Hot–Cold* scale presented the highest uncertainty in the perceptual assessment, and 31.4% of assessments in this scale were *Can’t Say*. This is reflected in median ratings, with 6 out of 11 materials being rated as *Can’t Say* on this scale. This was interesting as participants were touching the samples, yet they were unable to accurately classify the materials on perceived temperature, unlike perceived roughness. This may be due to smaller contrasts in the thermal conductivity of materials, which made it difficult for the participants to perceive fine temperature variations in comparison with their body temperature [43].

*Aged-New* had high uncertainty in responses (21.8%) and had the least number of significant correlations. These factors make the assessments from this attribute unreliable and may be used only as speculative. *Aged* exhibited significant correlations with the *Interesting–Boring* scale, showing a negative correlation between *New* and *Interesting* (Appendix A). This may be due to the presence of visual and tactile features that exist on the material samples that resemble ageing; these features invite users to interact with and explore the material.

To sum up the attribute–attribute correlations:Some attribute scales are closely interlinked with each other, such as complexity–interestingness–unusualness and naturality–beauty–value;Natural-rated materials are also rated highly for ordinariness, presumably due to the ease of visual processing and familiarity;Perception of value correlates with beauty, but it is not dependent solely on beauty, and value may also be influenced by the perceived smoothness of the material.

### 3.3. Combining Correlations

Combining the significant correlations observed in the bimodal study (visual + tactile) uncovers general correlations amongst attribute scales and further relationships amongst them (Appendix A). Some attribute pairs such as *Simple–Ordinary* and *Complex–Simple, Beautiful–Ugly* and *Valuable–Worthless,* and *Natural–Artificial* and *Valuable–Worthless*, illustrate strong correlations through inverse as well as antonymous correlations amongst these attributes. It is seen that complexity of the materials correlated with unusualness or novelty, as well as the interestingness of these materials. This relationship could be expressed as:*Complexity α Novelty α Interesting*(1)

Highly complex materials are likely to be rated highly for novelty and interestingness. This aspect of biobased composites is significant, as the same correlations were also observed with visual perception [22]. Hence, we could consider this perceptual dimension as a visually dominant one. The bimodal study also revealed the relationship between beauty, value, and naturality as expressed below:*Beauty α Worth α Naturality*(2)

While the relationship between beauty and value was observed earlier [22], this bimodal study unveiled the relationship between these attributes and naturality. These properties were consistently high for Cluster 2 materials, followed by UD Flax, Cellulose + Wood, and NW coir, all of which are materials in Cluster 4 as per t-SNE dimension reduction. This may indicate the dominance of visual aspects along with the bimodal appreciation of visual/tactile incongruity leading to perceptual surprise, as argued by Ludden et al. [44]. All the natural materials fit into the concept of ‘Rhymic’ visual character as classified in [25], whereas Cellulose + Wood and UD Flax fall in the random–rhymic region visually. However, the visual order concept cannot explain all the perceptual behaviour for naturality, as tactility also affected it. An example is Rayon, which falls in the same visual category as Cellulose + Wood but is classified differently in the beauty–naturality attributes due to its tactile characteristics.

The next attribute relationship for materials revealed was amongst *Artificial, Cold*, and *Strong*. Another analogues relationship which had a common factor was amongst *Smooth, Worth, Simple*, and *Strong*. These relationships are depicted as:*Artificial α Cold α Strong*(3)
*Smooth α Worth α Simple α Strong*(4)

These correlations were also detected during the visual perception study [25] and may be due to the distinct presence of resins on the material surface and the relative consistency in surface patterns. These relationships may be bimodal, with both the coldness associated with the tactility of the resin surface and the visual consistency of simple interfaces prompting higher quality and, thus, material strength. This also brings the relationship loop back to *Worth* in Equation (2) and thus to beauty and naturality. Another antonymous correlation is observed for the *Interesting–Boring* scale through the attribute *New*:*New α Boring*(5)

This could also have a link with Equation (1), and so more aged-looking interfaces would be considered more interesting. This may be attributed to the visual complexity of the aged surfaces, deviating from the monotony of new surfaces and the resulting correlation with interestingness. 

The core objective of this study has been to identify the attribute characteristics that form the perception of a natural, beautiful, and valuable material, and combining the five equations gives a general set of rules for a desirable biobased composite:The qualities of beauty, value, and naturality are correlated in biobased composites;Beauty is predominantly a visual but bimodal characteristic; complexity and novelty increase interestingness;Agedness may contribute to visual complexity, and this leads to interestingness;*Artificial* and *Cold* rated materials are evaluated as *Strong*. *Strong* materials are expected to have simple, consistent, and smooth surfaces and will have a higher value.

While the first three rules are congruent, the fourth rule for value conflicts with the requirements for beauty. This relationship is visualised in Figure 8. *Naturality, Beauty*, and *Value* are the three major components of an ideal biobased composite, and all of these are mutually correlated. However, naturality and beauty have a closer relationship, both being visually influenced attributes. These include visual subfactors such as complexity, interestingness, and novelty under *Beauty*; and material characteristics that correlate with these attributes have been noted in perception studies (visual and visual–tactile). It is seen that *Naturality* does not possess unique subattributes as in the case of beauty and value; rather, *Naturality* is tied with all these components through direct and indirect correlations. This may be because naturality is multidimensional [36], making it difficult to isolate its constituent attributes explicitly. 

The role of *Value* is interesting, as it draws relationships from *Beauty* and *Naturality* as well as a few other subcomponents. These may be broadly classified into *Aesthetic Value*—value from aesthetics (beauty and naturality) and *Functional Value*—value from functionality (such as strength and smoothness). It, however, presents antagonistic relationships between subcomponents (*Cold* versus *Natural, Simple* versus *Complex*) which leads to an inverse relationship between functional value and aesthetic value. This is very useful in creating new biobased materials, as focus may be put on either value based on the type of constraint (functional or aesthetic). For example, if the versatility of the visual aspects of the biobased composite is limited, it may be advisable for the manufacturers and designers to focus on the functional value components to improve material desirability and vice versa. 

While the attribute subcomponents of beauty, naturality, and value are defined, it is also important to identify the physical characteristic analogues of these attributes. What material properties—visual or tactile, create the perception of beauty, naturality, and value or their subcomponents? This may be deducted from correlating the attribute ratings and material characteristics from the visual and the bimodal study. The relationships deduced through the analysis of attribute–attribute correlations in this study and Manu et al. [25] point to various clues, as listed in Figure 8. 

In the case of *Beauty*, these include complex visual characteristics but with rhymic and organic patterns; the presence of contrasting fibres, but visually harmonious with the matrix materials; and simple transitions in colour and texture, eliminating confusion from complexity. In the case of *Naturality*, these factors include warm colours, organic patterns, and visibly fibrous but with surface textures. Visual harmony is important here as well, as naturality is enhanced in material interfaces, with both components (matrix and reinforcement) having a balanced visual impact. Materials with hard and smooth surfaces were generally assessed as more valuable, and ordered patterns signalled higher material strength and, thus, a higher value. However, it should be noted that such materials are generally weakly rated on naturality, requiring a balance of these characteristics for the ideal material perception. Visual inconsistencies indicate defects in materials and thus less value; at the same time, such inclusions add a layer of complexity and improve the perception of beauty. A way to achieve the balance of order–disorder, harmony–discord, and contrast–likeness in fibre reinforcements is to have partially ordered (e.g., unidirectional) fibres with a high length-to-diameter ratio. Focus may be given to these material characteristics for future biomaterial formulations to achieve the desired material perception.

## 4. Conclusions

Through this study, we aimed to examine the relationships between material perception and sensory characteristics of various biobased materials; the combination of results from the bimodal and the unimodal (visual) perception study reveals the key variables that may govern the formation of material perception. This study also provided an opportunity to compare the perception of materials in unimodal (visual only) and bimodal (visual + tactile) contexts, thereby being able to establish the influence of sensory stimuli on perception formation. It was noted that for tactilely dominant attributes such as roughness, visual perception may be highly inaccurate, and this requires closer attention from manufacturers and retailers who focus on digital marketing and retail channels. It is also observed that the components of perception need not be consistent along material categories and are uniquely dependent on their sensory characteristics. It also provides valuable hints to material and product designers about tactile perception.

Tactile components also influenced the assessments of naturality, and value, pointing towards a bimodal perception mechanism for these attributes. This study also revealed a set of attribute relationships that constitute the key qualities of beauty, naturality, and value, which also had antagonistic factors that need to be balanced. Comparing the underlying material characteristics from both perception studies, a set of principles for material characteristics are also suggested, which may help leverage individual perceptual qualities. This may help us to ‘design’ materials with specific physical features which might elicit the desired perception in their users [21]. This study also points to the fact that the assessment of variables such as naturality and desirability may not be quantitative and more of categorical nature, as suggested by Overvliet and Soto-Faraco [36]. It is also multidimensional, as illustrated in the component framework, and any attempts to ‘design’ perceptions must have a holistic look at those variables.

Further explorations using this approach of ‘perception-based design’ for creating novel materials may be highly relevant for product designers and manufacturers, enabling them to customise product perception to better suit the emotional and aesthetic demands of various consumer segments. This will help overcome perceptual handicaps about biobased composites and improve consumer–product attachment, thereby increasing product lifespan and reducing waste generation.

## Figures and Tables

**Figure 1 materials-16-01844-f001:**
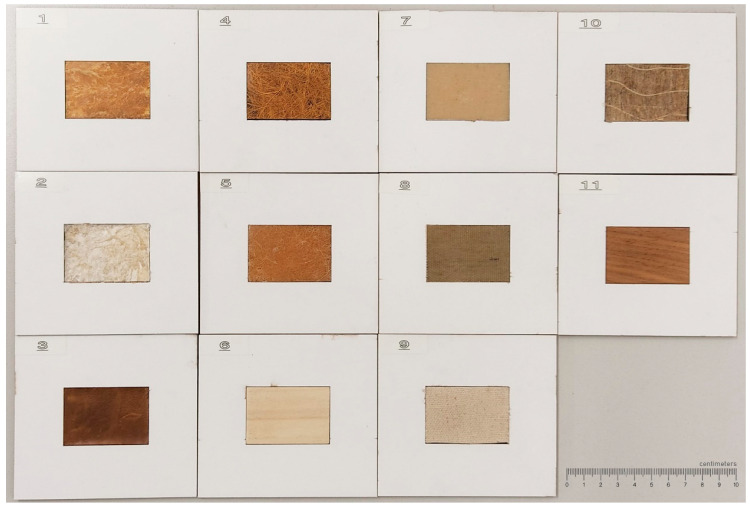
Image of material samples mounted on frames: (**1**) Cellulose + Wood, (**2**) Rayon, (**3**) Leather, (**4**) NW Coir, (**5**) NW Sisal, (**6**) Poplar, (**7**) TW Cotton, (**8**) TW Flax 1, (**9**) TW Flax 2, (**10**) UD Flax, and (**11**) Walnut.

**Figure 2 materials-16-01844-f002:**
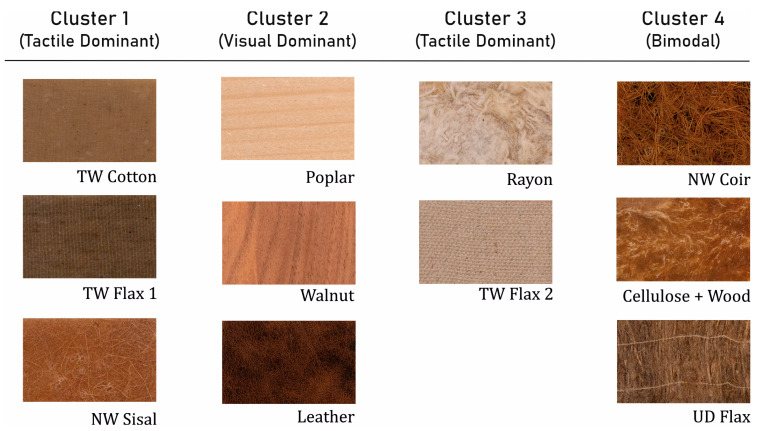
Material clusters and dominant sensory channels.

**Figure 3 materials-16-01844-f003:**
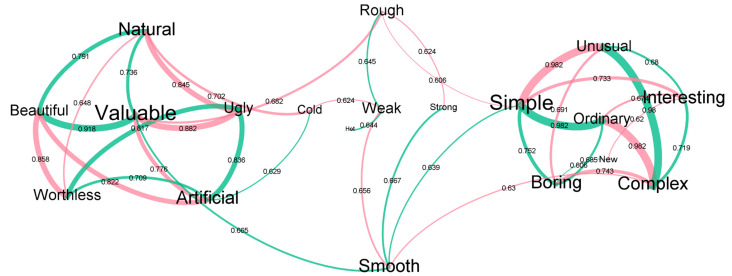
Significant correlations amongst various attributes. The text sizes of attributes are proportional to the degrees of correlation with other attributes. Green lines indicate positive correlations, and the red line indicates negative correlations. The thickness of lines is proportional to Spearman’s correlation coefficient between the attributes.

**Figure 4 materials-16-01844-f004:**

Materials with high naturality rating.

**Figure 5 materials-16-01844-f005:**
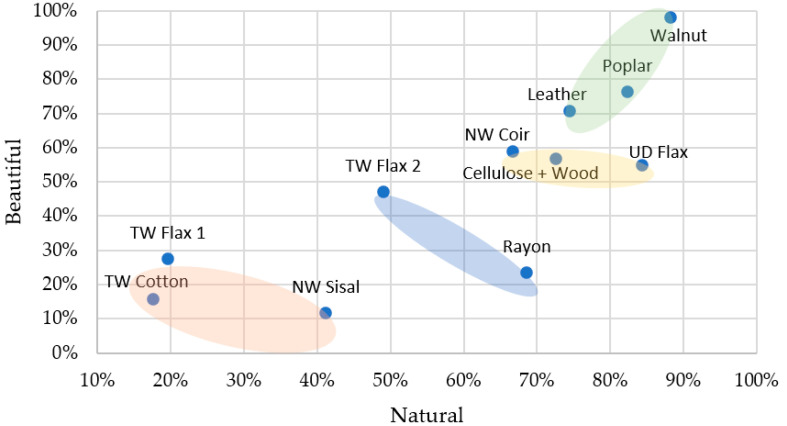
Material perception for Natural–Beautiful (visual + tactile perception). ■ Cluster 1; ■ Cluster 2; ■ Cluster 3; ■ Cluster 4.

**Figure 6 materials-16-01844-f006:**
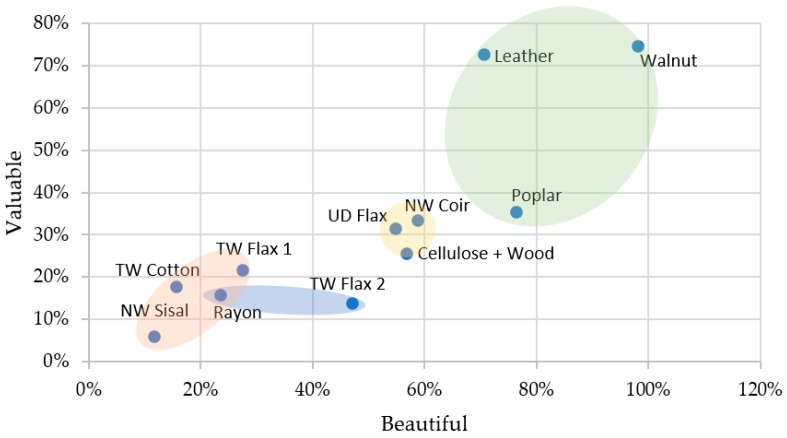
Material perception for Beautiful–Valuable (visual + tactile perception). ■ Cluster 1; ■ Cluster 2; ■ Cluster 3; ■ Cluster 4.

**Figure 7 materials-16-01844-f007:**
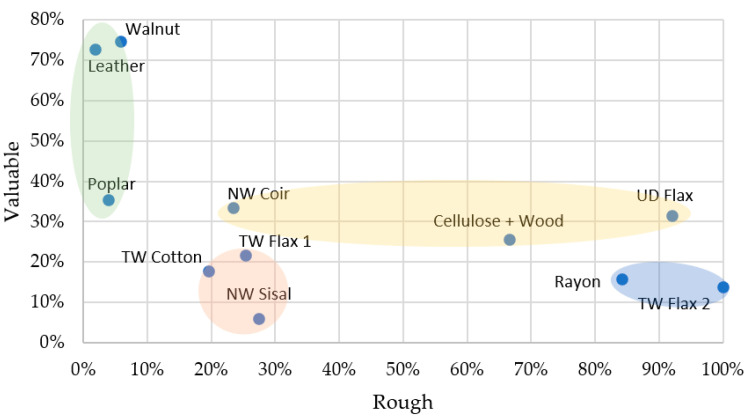
Material perception for Valuable–Rough (visual + tactile perception). ■ Cluster 1; ■ Cluster 2; ■ Cluster 3; ■ Cluster 4.

**Figure 8 materials-16-01844-f008:**
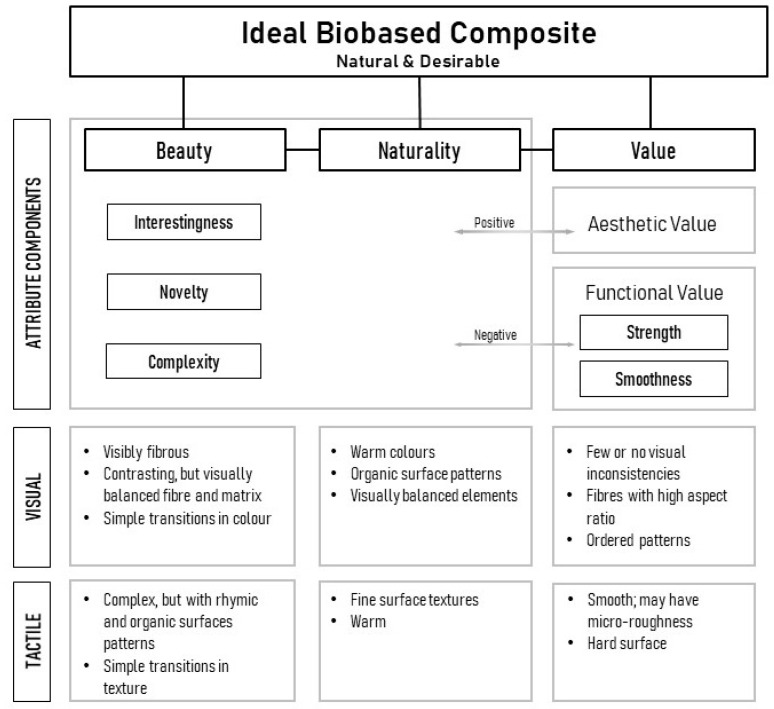
Components and internal relationships for naturality and desirability.

**Table 1 materials-16-01844-t001:** List of attributes for the bimodal study (visual + tactile).

Attributes (Adjective Pairs)
Aged–New
Complex–Simple
Interesting–Boring
Natural–Artificial
Unusual–Ordinary
Beautiful–Ugly
Valuable–Worthless
Strong–Weak
Rough–Smooth
Hot–Cold

## Data Availability

Not applicable.

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
