# Peer review of "Visual–Tactile Perception of Biobased Composites"

_materials, 2023, doi:10.3390/ma16051844_

Round 1

Reviewer 1 Report

Dear Authors thanls for your work. The paper needs same correction and revision.

Summary should be developed. Brief information can be given about the emergence of the study, the problem, the solution method and its results.

References should be noted. It should be given as [x1-x11] or [x1,x2]. Reproducibility must be ensured.

10-11-12-13-14-15-18-19-20

Attention should be paid to the order of sources. [4]

Commercial company names may cause conflict of interest. Brand-Company name should be avoided as much as possible.

Author Response

Dear Authors thanls for your work. The paper needs same correction and revision.

Summary should be developed. Brief information can be given about the emergence of the study, the problem, the solution method and its results.

Modifications made. Brief information about the context of the study, the objectives of the study, the materials and methods are provided. Additional details have been added in the Supplementary Data document. A more detailed discussion of the context has been done in the following cited references from the authors:

  1. Manu, A. R. Nazmi, B. Shahri, N. Emerson, and T. Huber, “Biocomposites: A Review of Materials and Perception,” Mater. Today Commun., p. 103308, 2022, doi: 10.1016/j.mtcomm.2022.103308.
  2. Manu, A. R. Nazmi, B. Shahri, N. Emerson, J. Müssig, and T. Huber, “Designing With Biobased Composites: Understanding Material Perception Through Semiotic Attributes,” 2022, doi: 10.31219/OSF.IO/7TS2X.

References should be noted. It should be given as [x1-x11] or [x1,x2]. Reproducibility must be ensured.

10-11-12-13-14-15-18-19-20

Attention should be paid to the order of sources. [4]

In-text citation format modified as suggested.

The case of some sources seeming out of place is when a previously referred source is cited again later in the manuscript.

Commercial company names may cause conflict of interest. Brand-Company name should be avoided as much as possible.

All brand names were removed and replaced with generic material names (e.g., Cordenka changed to Rayon)

Reviewer 2 Report

Dear Authors,

This manuscript is really very interesting and I think it is needed. However, one should take a slightly broader look at the issues described in the introduction.

Detailed coments below.

Line 35: What do you mean by low attractiveness? It is worth presenting here the typical features of bicomposites, e.g. dark color, visible structure, visible plant fibers, etc. Generally, in many works you can see photos of various biocomposites. Check out the following article: "Properties of biocomposites from rapeseed meal, fruit pomace and microcrystalline cellulose made by press pressing: Mechanical and physicochemical characteristics", "A review on alternative raw materials for sustainable production: novel plant fibers".

Line 40: There remains one more important sense, "smell". You haven't studied this sense, but it's worth mentioning in the context of broader research. For example, biocomposites (since they are often made from plant-based raw materials) may also have an odor (fragrant/neutral/or unpleasant). Such an example can be biocomposites made of fermented plant particles, e.g. "Properties of biocomposites produced with thermoplastic starch and digestate: Physicochemical and mechanical characteristics". Generally, however, most biocomposites may have a characteristic smell. This shows how complex the process of introducing such new materials to local markets can be.

Line 90: I think you should justify why you want to do this research. Why are important. Define the scientific problem precisely.

Line 14: Add what software you used to analyze the data, name: manufacturer, city, country.

Line 448: Write one more forward-looking request. What can change in the design of new/innovative biocomposites in the context of your research?

Author Response

Dear Authors,

This manuscript is really very interesting and I think it is needed. However, one should take a slightly broader look at the issues described in the introduction.

Detailed coments below.

Line 35: What do you mean by low attractiveness? It is worth presenting here the typical features of bicomposites, e.g. dark color, visible structure, visible plant fibers, etc. Generally, in many works you can see photos of various biocomposites. Check out the following article: "Properties of biocomposites from rapeseed meal, fruit pomace and microcrystalline cellulose made by press pressing: Mechanical and physicochemical characteristics", "A review on alternative raw materials for sustainable production: novel plant fibers".

Modified as suggested (Lines 36-38)

Line 40: There remains one more important sense, "smell". You haven't studied this sense, but it's worth mentioning in the context of broader research. For example, biocomposites (since they are often made from plant-based raw materials) may also have an odor (fragrant/neutral/or unpleasant). Such an example can be biocomposites made of fermented plant particles, e.g. "Properties of biocomposites produced with thermoplastic starch and digestate: Physicochemical and mechanical characteristics". Generally, however, most biocomposites may have a characteristic smell. This shows how complex the process of introducing such new materials to local markets can be.

The approach of this study is in the context of product design and hence focus is placed on vision and tactility which are the dominant senses. Modification made through lines 46-47 and 114-115.

Line 90: I think you should justify why you want to do this research. Why are important. Define the scientific problem precisely.

Clarification added (Lines 99-100). A detailed discussion of this topic has been done in cited reference: T. Manu, A. R. Nazmi, B. Shahri, N. Emerson, and T. Huber, “Biocomposites: A Review of Materials and Perception,” Mater. Today Commun., p. 103308, 2022, doi: 10.1016/j.mtcomm.2022.103308.

 Line 14: Add what software you used to analyze the data, name: manufacturer, city, country.

Clarification added (Lines 135-137, 161-163)

Line 448: Write one more forward-looking request. What can change in the design of new/innovative biocomposites in the context of your research?

Clarification added (Lines 484-489)

Reviewer 3 Report

the article is well-written and all the relevant data are provided in the supplementary file. Minor changes are suggested in the attached file. 

Author Response

please consider adding data of a control material sample of which the perception is well known or to point out why a control material was not used

Clarification added (Lines 112-114)

please state if the study was evaluated by the ethical committee of your institution

This study was approved by the Human Ethics Committee of the University of Canterbury, Christchurch, New Zealand (HEC 2021/22/LR-PS dated 04.06.2021) (Lines 500-501)

please, for the sake of clarity, consider to stress out the concerns about the Semantic Differential method

(e.g.Although the semantic differential technique has been widely used since its inception, there are some concerns about it. Theoretically, using the scale rests on the assumption that humans' connotations of a word are not the same, which is why this technique is needed. Paradoxically, the scale assumes that the chosen adjectives mean the same to everyone. For the scale to work, it must be assumed that humans share the same connotations for some words (i.e. the bipolar adjectives). This is a fairly strong assumption. For instance, looking at Table 1, the bipolar pairs of adjectives generally are set out with "positive" adjectives on the left. However, for the cold/hot dyad it is not clear that "cold" is always associated with positive thoughts. Indeed, depending upon the respondent's past, "cold" could easily evoke negative thoughts.

Another concern is that attitudes do not always match up with behavior. As such, attitudes can be poor predictors of action or behavior. The semantic differential ought to be able to overcome these concerns simply by design, but that does not mean that it can do so on its own. If respondents give socially desirable answers, it will negatively impact the reliability of the measure. Also, if a respondent begins to consistently answer in the same way (i.e. all neutral or always agreeing), the reliability must be questioned. Yet another critique is that the semantic differential does not actually identify individual emotions. Because of its design, the semantic differential technique cannot distinguish beyond the one continuum, but then it never was intended to do so in the first place.)

Clarification added (Lines 137-142)

please consider adding a reference

Reference added (L. van der Maaten and G. Hinton, “Visualizing data using t-SNE.,” J. Mach. Learn. Res., vol. 9, no. 11, pp. 2579–2605, 2008)

Reviewer 4 Report

Dear authors,

We have read your manuscript describing the experiments conducted on volunteers, to assess their perception about bio based materials.

The topic is unusual, with an interesting approach. But, as it is unusual, the whole study should be accompanied with additional explanations, in order to justify the consistency and relevance of the results presented.

> The Results are clearly described, and well analyzed.

> The Methods are clear, but we miss some informations : we'd be interested to know on which criteria you have chosen the various materials; it'd be also interesting to know how the volunteers were recruited (could there be a bias in the population? i.e only people working in the field of Materials-Chemistry-Physics...)

> Our major concern is about the 'Introduction' section. A description of the fields of application should be presented, with explanations about the need of sustainable composites in these specific fields (what are the materials used? Why composite materials are they needed? What are the elements that hinder the use of sustainable composites?...)

> In the same idea, the conclusion should create a link between the fields of application (to be described in 'Introduction') and the results obtained in this study.

Therefore we ask for modification before acceptance of your manuscript.

Author Response

Dear authors, We have read your manuscript describing the experiments conducted on volunteers, to assess their perception about bio based materials.

The topic is unusual, with an interesting approach. But, as it is unusual, the whole study should be accompanied with additional explanations, in order to justify the consistency and relevance of the results presented.

> The Results are clearly described, and well analyzed.

> The Methods are clear, but we miss some informations : we'd be interested to know on which criteria you have chosen the various materials; it'd be also interesting to know how the volunteers were recruited (could there be a bias in the population? i.e only people working in the field of Materials-Chemistry-Physics...)

Clarification added (Lines 103-105, 120-124)

> Our major concern is about the 'Introduction' section. A description of the fields of application should be presented, with explanations about the need of sustainable composites in these specific fields (what are the materials used? Why composite materials are they needed? What are the elements that hinder the use of sustainable composites?...)

Modifications made.  Please see revised manuscript for details. 

> In the same idea, the conclusion should create a link between the fields of application (to be described in 'Introduction') and the results obtained in this study. Therefore we ask for modification before acceptance of your manuscript.

Modifications made. Please see revised manuscript for details. 

Round 2

Reviewer 1 Report

Accept it with this form

Reviewer 2 Report

Dear Authors,

I accepts the corrections made.